# SLR, GRACE and Swarm Gravity Field Determination and Combination

**Ulrich Meyer** [1,*]**, Krzysztof Sosnica** [2] **, Daniel Arnold** [1]**, Christoph Dahle** [1,3] **, Daniela Thaller** [4]**, Rolf Dach** [1] **and Adrian Jäggi** [1]

1   Astronomical Institute, University of Bern, 3012 Bern, Switzerland; daniel.arnold@aiub.unibe.ch (D.A.); dahle@gfz-potsdam.de (C.D.); rolf.dach@aiub.unibe.ch (R.D.); adrian.jaeggi@aiub.unibe.ch (A.J.)
2   Institute of Geodesy and Geoinformatics, Wroclaw University of Environmental and Life Sciences, 50-375 Wroclaw, Poland; krzysztof.sosnica@igig.up.wroc.pl
3   GFZ German Research Centre for Geosciences, 14473 Potsdam, Germany
4   Bundesamt für Kartographie und Geodäsie, 60598 Frankfurt, Germany; daniela.thaller@bkg.bund.de
*   Correspondence: ulrich.meyer@aiub.unibe.ch

**Abstract:** Satellite gravimetry allows for determining large scale mass transport in the system Earth and to quantify ice mass change in polar regions. We provide, evaluate and compare a long time-series of monthly gravity field solutions derived either by satellite laser ranging (SLR) to geodetic satellites, by GPS and K-band observations of the GRACE mission, or by GPS observations of the three Swarm satellites. While GRACE provides gravity signal at the highest spatial resolution, SLR sheds light on mass transport in polar regions at larger scales also in the pre- and post-GRACE era. To bridge the gap between GRACE and GRACE Follow-On, we also derive monthly gravity fields using Swarm data and perform a combination with SLR. To correctly take all correlations into account, this combination is performed on the normal equation level. Validating the Swarm/SLR combination against GRACE during the overlapping period January 2015 to June 2016, the best fit is achieved when down-weighting Swarm compared to the weights determined by variance component estimation. While between 2014 and 2017 SLR alone slightly overestimates mass loss in Greenland compared to GRACE, the combined gravity fields match significantly better in the overlapping time period and the RMS of the differences is reduced by almost 100 Gt. After 2017, both SLR and Swarm indicate moderate mass gain in Greenland.

**Keywords:** satellite gravimetry; ice mass change; GRACE; SLR; swarm; normal equation combination

## 1. Introduction

The Gravity Recovery and Climate Experiment satellite mission (GRACE) [1] dedicated to the observation of temporal variations of the gravity field allows for the quantification of ice mass loss of glacier accumulations in polar and sub-polar regions (e.g., [2–4]). However, this high resolution information is limited to the life-time of the GRACE satellites (2002–2017) and of the GRACE-FO (Follow On) [5] mission that was launched in May 2018, but suffered a failure of the main instrument processing unit between July and October 2018. To date, no other single satellite mission or proxy is able to bridge the gap between GRACE and GRACE-FO with comparable quality (compare, e.g., [6,7] or [8]). We therefore present a combination on the normal equation level of alternative satellite gravimetric data considering several missions.

Satellites not dedicated to gravity field determination already collected data before GRACE and during the gap between GRACE and GRACE-FO. These are mainly the geodetic satellite laser ranging (SLR) missions, e.g., the Laser Geodynamics Satellites (LAGEOS) or, at a significantly

lower orbit altitude, the Laser Relativity Satellite (LARES) as the youngest member of the SLR family, which also provide information about temporal variations of the Earth gravity field at the lowest degrees of the spherical harmonic spectrum. The geodetic SLR satellites are optimal for gravimetry by their spherical geometry and their favorable area-to-mass ratio that minimizes the effect of surface forces [9]. For studies of temporal gravity field variations derived by SLR in the pre-GRACE era, see Cheng et al. [10], Bianco et al. [11] and Cheng and Tapley [12]. For a focus on variations in Earth oblateness, where GRACE results are unreliable [13], compare Cox and Chao [14] Cheng and Tapley [15] and Bloßfeld et al. [16].

In addition, other Earth observing satellites at orbits below 1500 km altitude, so-called low Earth orbiters (LEOs), which are equipped with GPS receivers for precise orbit determination, may serve for deriving the mass distribution of the Earth and its temporal variations at low to medium resolution [17]. Due to the time-period covered and their low orbit altitude, the three satellites of the Swarm mission [18] are well suited to bridge the gap between GRACE and GRACE-FO.

We present a long-term series covering 1995–2018 of low resolution monthly gravity field coefficients derived at the Astronomical Institute of the University of Bern (AIUB) from a combined orbit determination of LAGEOS and the geodetic SLR satellites in low Earth orbits [19]. The derived gravity fields are compared to time-series of GRACE solutions spanning 2003–2016, which were also determined at AIUB [20]. Finally, also the GPS-derived Swarm kinematic orbits that are routinely computed at AIUB [21] are exploited to derive monthly gravity fields for the time-period 2014–2018.

Due to the different orbital altitudes and ground track patterns of the satellites, and due to the different observation techniques used, the gravity fields derived differ in their spherical harmonic and corresponding spatial resolution [22]. The truncation of the spherical harmonic expansion of the gravity field leads to spatial leakage that has to be taken into account when comparing the results (e.g., [4,23–26]). Furthermore, the high-resolution GRACE gravity fields commonly are smoothed to suppress noise in the high-degree coefficients (e.g., [27–29]). We shortly introduce the representation of the gravity field in a spherical harmonic expansion and exemplify the problem of signal loss due to leakage or filter attenuation in Section 2. In the following, all monthly gravity fields are truncated to the same spherical harmonic degree for the sake of comparison of the different techniques.

To illustrate the capability of the different satellite gravimetric techniques to track temporal variations of the mass distribution within the system Earth, we transform the gravity variations to variations in equivalent water height (EWH) [30] or mass variations in Greenland and at the coast of West Antarctica and compare their spatial distribution and development with time. Moreover, we fit deterministic signal models of secular and seasonal variations to consecutive five-year time-periods and compare the derived mass trends.

Comparable studies were already performed by Matsuo et al. [31], Talpe et al. [32] and Bonin et al. [33]. The latter concluded that truncated at spherical harmonic degree 5 gravity field models are not able to correctly separate mass loss in Greenland and Antarctica and that, while the inter-annual variations in none of the SLR time-series are realistic, long-time mass trends are well captured in the regularized AIUB SLR time-series truncated at degree 10 [19] as opposed to other series. Bonin et al. [33] suggest to either reduce the temporal resolution of the SLR-derived gravity fields to reduce their scatter, which in our eyes is counter-intuitive because it also reduces the separability of mass loss signal in Greenland and Antarctica, or to combine SLR with other data.

We do not continue our regularized SLR-only degree 10 time-series beyond 2014 because in combination with Swarm the full sensitivity of SLR can be exploited without regularization. The combination approach is favorable because it is independent from external information not based on the original observations. We provide an unconstrained SLR-only degree 6 time-series and a combined SLR + Swarm time-series where SLR entered to degree and order 10. The decorrelation of the individual spherical harmonic coefficients is possible by the Swarm data. Due to the consistent processing of Swarm and SLR, we are able to perform the combination on the normal equation level, taking correlations between reference frame (co-estimated in the case of SLR), force model and

orbit parameters into account [34]. Combinations of SLR with CHAllenging Minisatellite Payload (CHAMP), GRACE or Gravity field and steady-state Ocean Circulation Explorer (GOCE) for gravity field determination were also studied by Moore et al. [35], Cheng et al. [36], Maier et al. [37] and Haberkorn et al. [38].

## 2. Materials and Methods

The dominating force acting on a satellite is due to Earth gravity. Satellite orbits at low altitudes are sensitive to mass distribution and redistribution and therefore to mass transport in the Earth system. The long-term mean gravity field of the Earth has been determined by the dedicated gravimetric satellite mission GFZ-1 [39], by CHAMP [40] that also observed the magnetosphere, and by GRACE [41,42] and GOCE [43,44], again dedicated to gravimetry. Temporal variations in gravity have mainly been derived from GRACE (see Wouters et al. [45] for an overview), but also from CHAMP [46] and other Earth observing LEOs such as the three satellites of the Swarm mission [47] or from the fleet of geodetic SLR satellites (e.g., [10–12]).

Prerequisite for gravity field determination is the precise observation of the satellite orbits. Today this may either happen by laser ranging or by GPS- or Doris-tracking of the satellites (the latter being irrelevant here due to the choice of missions). Critical for orbit modeling and signal separation is the knowledge of all forces acting on the satellites, the so-called background force model, and the set of parameters estimated to represent the orbit, improve the a priori force model and absorb model or instrument errors, the so-called orbit parameterization. In detail, the force model consists of a priori models of the gravitational forces by the Earth and third bodies, ocean, atmosphere and solid Earth tides, satellite specific models of the surface forces such as air drag, solar radiation pressure and Earth albedo, and an empirical or pseudo-stochastic part [48]. In case of dedicated gravity missions, the surface forces are normally measured by accelerometers onboard the satellites [49] (due to technical reasons only the sum of all forces plus an unknown but constant bias can be observed).

At AIUB, the gravity field determination from satellite observations is treated as a generalized orbit determination problem [48]. The satellites' orbits are co-estimated with the instrument specific parameters like accelerometer scale factors and with the parameters updating the a priori background forces, i.e., the weight coefficients $C_{lm}$ and $S_{lm}$ of the expansion of the Earth's gravitational potential $V$ in spherical harmonics [50]

$$V(r,\phi,\lambda) = \frac{GM}{R} \sum_{l=0}^{\infty} \left(\frac{R}{r}\right)^{l+1} \sum_{m=0}^{l} P_{lm}(\sin\phi)\left(C_{lm}\cos m\lambda + S_{lm}\sin m\lambda\right), \tag{1}$$

where $r, \phi, \lambda$ are the spherical coordinates in an Earth-fixed reference frame, $GM$ is the product of the gravitational constant and the Earth's mass, $R$ is the semi-major axis of the Earth and $P_{lm}$ are the fully normalized Legendre functions of degree $l$ and order $m$. In case of SLR, air drag scale factors, station and geocenter coordinates, and Earth orientation parameters are also co-estimated [19].

Deficiencies of the force model are mitigated by pseudo-stochastic accelerations or pulses [51]. In order to reduce the absorption of gravity field signal by these parameters, their frequency has to be tailored to the orbit altitude and observation sampling of the specific satellite and their magnitude has to be limited by constraints depending on the satellites' environment at orbit altitude. By extensive cross-validation with other analysis centers in the frame of the European Gravity Service for Improved Emergency Management (EGSIEM) [52] project, it could be shown that mass trends and the amplitude of seasonal variations in the AIUB gravity fields are not affected by the pseudo-stochastic parameterization. Since a separate determination of sub-groups of the parameter space leads to a regularization favoring the a priori force model applied [53] orbit, stochastic and force model parameters have to be determined together in one adjustment process.

The observations used for orbit determination are either the 1D-ranges (normal points [54]) observed by SLR, the kinematic orbits of Earth observing LEOs determined by precise point

positioning (PPP) [55] from high-low GPS data that are used as pseudo-observations [56], or low-low range-rates derived from the K-band inter-satellite link in case of the two satellites of the GRACE mission [57]. The sampling of these observation types is very diverse. In case of SLR, it depends on the inhomogeneous global distribution of the SLR stations [19]. Certain regions of the Earth, such as the polar regions, are not covered by observations at all due to the lack of suitably positioned stations. All information about gravity variations is derived from the orbit dynamics. Therefore, the orbit modeling of SLR satellites has to be mainly dynamical (based on physical models) and the pseudo-stochastic parameterization of the orbits has to be very limited to allow for gravity field determination. On the other hand, GPS and K-band observations are normally given at very high sampling rates of 1 s, 5 s or 10 s and in case of polar orbits the observation distribution is global and densest near the poles. Several pseudo-stochastic parameters may be set up per orbital revolution of the Swarm or GRACE satellites.

The spatial resolution of the resulting gravity fields depends on the orbital altitude and ground track pattern of the satellites. In case of sparse SLR tracking, gravity variations beyond degree 2 can only be determined by the combined evaluation of several satellites (see [10–12]). Satellites at lower inclinations are helpful to decorrelate the individual gravity field coefficients [19]. In case of the Earth observing LEOs, the achievable temporal resolution of subsequent gravity field solutions directly depends on sub-cycles of the ground track pattern, which vary depending on the orbit altitude (decreasing with time in case of the LEOs). The GRACE mission was designed to deliver monthly gravity fields [1] and since then monthly temporal resolution has become the standard, even if 10-day [58], and even daily "snapshot" solutions [59] of the gravity field are also available.

## 2.1. SLR

An important prerequisite for the determination of the long wavelength part of the gravity field is the exact modeling of the surface forces acting on the satellites. To simplify the modeling of the surface forces, most of the geodetic SLR satellites have a low area-to-mass ratio [9]. The high-flying SLR satellites LAGEOS 1 and 2 are mainly sensitive to the Earth's flattening $C_{20}$ and its variations with time [60]. Higher spectral resolution of the gravity field can be achieved by combined processing of the fleet of geodetic SLR satellites at lower orbits (SLR-LEOs) and at different inclinations of the orbital plane (see [10–12,19]). We exploit the two LAGEOS satellites and the following SLR-LEOs for the determination of large scale temporal gravity field variations: Starlette, AJISAI, Stella, Larets, LARES, and the old Earth observation satellite Beacon-C that is carrying a laser retro-reflector array and is included due to its low orbit inclination. The orbit characteristics of all satellites used are compiled in Table 1.

**Table 1.** Orbit characteristics of satellites used and a priori observation errors, as determined by variance analysis of residuals (Beacon-C is down-weighted due to large surface forces acting on the non-spherical satellite).

| Satellite | Launch Date | Orbit Altitude | Inclination | Observation Type | A Priori Error |
|---|---|---|---|---|---|
| Beacon-C | 1965 | 940–1300 km | 41.23° | SLR | 50 mm |
| Starlette | 1975 | 800–1100 km | 49.84° | SLR | 20 mm |
| LAGEOS-1 | 1976 | 5860 km | 109.90° | SLR | 8 mm |
| AJISAI | 1986 | 1500 km | 50.04° | SLR | 25 mm |
| LAGEOS-2 | 1992 | 5620 km | 52.67° | SLR | 8 mm |
| Stella | 1993 | 810 km | 98.57° | SLR | 20 mm |
| Larets | 2003 | 690 km | 97.77° | SLR | 30 mm |
| LARES | 2012 | 1440 km | 69.56° | SLR | 15 mm |
| GRACE A and B | 2002 | 500 km in 03/2002 | 89° | GPS | 2 mm |
| | | | | K-band | 0.3 $\mu$m/s$^2$ |
| Swarm A and C | 2013 | 460 km in 04/2014 | 87.4° | GPS | 2–3 mm |
| Swarm B | 2013 | 530 km in 04/2014 | 88.0° | GPS | 2–3 mm |

The same a priori model of gravitational forces is consistently used for the SLR, GRACE and Swarm processing. To avoid affecting the derived temporal gravity field variations by a priori information, we use a static a priori gravity field. Furthermore, the background force model consists of solid Earth tides, ocean tides, ocean pole tides, and de-aliasing of ocean and atmosphere mass variations (AOD) [61]. Specific for the SLR satellites are models of the surface forces for air drag, solar radiation pressure and albedo that take into account the properties of the individual SLR satellites. The force model constituents and the resolution of their expansion in spherical harmonics (if applicable) are listed in Table 2.

**Table 2.** Background force model details for processing of low Earth orbiters. Where the force model constituent is expanded in spherical harmonics, the max. degree/order is given. Where this is not the case, it is only indicated if the listed model is applied or not. ACC indicates that surface forces are observed by accelerometers.

| Force | Model | SLR | GRACE | Swarm |
|---|---|---|---|---|
| Earth gravity | AIUB-GRACE03S (static) | degree/order 90 | 160 | 160 |
| Ocean tides | EOT11A [62] | degree/order 30 | 100 | 100 |
| Solid Earth tides | IERS2000 (elastic) [63] | yes | yes | yes |
| Ocean pole tide | Desai [63] | degree/order 100 | 100 | 100 |
| Atmosphere and ocean de-aliasing | RL05, RL06 (since 11/2017) | degree/order 100 | 100 | 100 |
| Air drag | NRLMSIS-00 [64] | yes | ACC | no |
| Solar pressure | uniform sphere approximation | yes | ACC | no |
| Albedo | Knocke [65,66] | yes | ACC | no |

Even if SLR is sensitive to gravity field variations at higher degrees, at monthly resolution, only spherical harmonics coefficients (SHC) of degrees 2 to 5 and of degree 6 and order 1 can be determined from SLR alone, i.e., without regularization. At higher degrees SLR-only gravity field solutions suffer from the strong correlations between individual SHC [19]. A prerequisite for the separation of individual SHC, even at the low degree of 5, is that SLR LEOs orbiting at different inclinations are used [10]. In this context Beacon-C and LARES are helpful due to their rather exotic orbit inclinations (Table 1).

However, the correct localization of the mass loss signal is not possible at this low resolution [33]. Therefore, we perform a combined solution with Swarm (see Section 2.4). In combination with Swarm, the SHC are decorrelated by the Swarm observations. To exploit the sensitivity of SLR beyond degree 5 in the combination and to avoid omission or commissioning errors [67], the SHC are set up to degree and order 10. In case of SLR-only monthly solutions coefficients of degree 6 (all but order 1), and degrees 7–10 are fixed to zero. In case of combined solutions, all coefficients are estimated.

The parameter-space of the SLR solutions is complemented by monthly estimates of the SLR station coordinates, by daily piecewise-linear estimates of the geocenter coordinates and by monthly estimates of the Earth rotation parameters [68]. As indicated above, the number of pseudo-stochastic parameters is very limited due to the sparse observation coverage. Only in along-track pulses are estimated once per orbital revolution of the satellites. The stochastic orbit parameterization is completed by periodic terms on orbit revolution frequency. The complete SLR orbit parameterization and the time intervals for which the individual parameters are set up are detailed in Table 3.

**Table 3.** Orbit parameterization of SLR satellites and time intervals for which the individual parameters are estimated.

| Parameter | LAGEOS-1/2 | SLR LEOs |
|---|---|---|
| Station coordinates | 30 days | 30 days |
| Earth orientation parameters | piecewise-linear, daily | piecewise-linear, daily |
| Geocenter coordinates | 30 days | 30 days |
| Gravity field | degree/order 10 | degree/order 10 |
| Range biases | selected sites | all sites |
| Initial state | 10 days | 1 day |
| Const. acceleration along-track | 10 days | none (air drag modelled) |
| Air drag scaling factor | none | 1 day |
| Periodic along-track | 10 days | 1 day |
| Periodic cross-track | none | 1 day |
| Pseudo-stochastic pulses | none | once per revolution along-track |

## 2.2. GRACE

The GRACE satellites were launched in March 2002 into polar orbits at an initial orbit altitude of 500 km [1]. During 15 years, they provided information about temporal variations of the Earth gravity field at unprecedented spatial and temporal resolution. The mission ended in October 2017 due to battery failure after decay to an orbit altitude of 330 km. The satellites were equipped with GPS receivers for orbit determination [57]. The key instrument for gravity field recovery was a K-band inter-satellite link which provided range measurements with micrometer accuracy [69]. Due to long-periodic systematic errors that cancel out by differentiation [70], most analysis centers use range-rates for gravity field recovery that were derived from the original range observations by numerical differentiation. Surface forces acting on the GRACE satellites were measured with onboard accelerometers in order to separate them from the gravitational forces [49].

We estimated monthly gravity fields from K-band range-rates and kinematic orbits that were used instead of the original GPS phase observations for efficiency reasons (the equivalence of both approaches was demonstrated by Jäggi et al. [71]). Therefore, in the first step, kinematic satellite positions were determined using the Bernese GNSS Software 5.2 (AIUB, Bern, Switzerland) [72]. The kinematic orbits were introduced as pseudo-observations together with their epoch-wise covariances [56]. Normal equations (NEQs) were computed for both observation types on a daily basis, combined and summed up to monthly batches. All but the gravity field parameters were pre-eliminated from the combined daily NEQs. Note that the K-band observations only contain line-of-sight, i.e., predominantly along-track information and a K-band only orbit solution therefore would be rank-deficit [70].

The accelerometer provides biased accelerations in the instrument frame that is closely aligned to the radial, along-track, and cross-track direction of the co-rotating orbital tripod. To account for the biases, daily constant accelerations are estimated in radial and cross-track directions. In along-track, four polynomial parameters are estimated per day to also absorb temperature induced variations in the accelerometer measurements that are in conflict with the ultra-sensitive K-band observations [20]. However, the key element of the Celestial Mechanics Approach (CMA) that was developed at AIUB for orbit and gravity field determination [48] is the estimation of rather frequent pseudo-stochastic accelerations (or pulses) to compensate deficiencies in the a priori force model [51]. Accelerations are estimated at 15 min intervals in all three directions of the orbital tripod. They are constrained to zero in order to minimize absorption of gravity signal. The parameterization applied for GRACE processing is summarized in Table 4.

For the a priori force model, the static part of AIUB-GRACE03S is used, but compared to the SLR-processing with increased spherical harmonic resolution to take into account the high sensitivity of the K-band observable. In addition, the background models for tidal effects and AOD were chosen consistently with the SLR-processing (details on the force model can be found in Table 2).

*2.3. Swarm*

A number of Earth observation LEOs at orbit altitudes of 400–600 km that are equipped with GPS receivers may also be exploited to derive information on gravity field variations [17,56]. Eminent candidates to bridge the gap, be it at lower spatial resolution, between GRACE and the GRACE-FO mission are the three satellites of ESA's Earth's Magnetic Field and Environment Explorer (Swarm) mission [18] that were launched in November 2013. All three satellites of the constellation circle the Earth in near circular polar orbits, Swarm A and C at 460 km, Swarm B at 530 km altitude (in 04/2014). They are equipped with GPS receivers and accelerometers, but the data of the latter turned out to be disturbed by slow temperature-induced bias variations and sudden bias changes [73] and are not routinely used for orbit and gravity field determination.

At AIUB, kinematic orbits of all three Swarm satellites are determined routinely [21]. While for the first mission phase until June 2014 gravity field results are deteriorated due to high solar and consequently ionosphere activity and not optimal GPS receiver settings [74,75], for the time period critical to bridge the gap between GRACE and GRACE-FO, the data quality corresponds to the nominal. By evaluation of the kinematic Swarm orbits, monthly gravity fields can be determined that are sensitive to temporal gravity variations up to about degree 13 [47].

The background force model for the Swarm processing is defined correspondingly to SLR and GRACE (see Table 2). Again the static part of AIUB-GRACE03S is used as a priori model and AOD is applied for de-aliasing of short-term variations in the atmosphere and ocean masses. As in the case of GRACE, no models for surface forces are applied. To compensate for the not used accelerometer observations, the constraints on the pseudo-stochastic accelerations are set less strict [21]. Details on the Swarm orbit parameterization can be extracted from Table 4.

**Table 4.** Orbit parameterization of Earth observation satellites and time intervals for which the individual parameters are estimated.

| Parameter | GRACE | Swarm |
|---|---|---|
| Gravity field | degree/order 60 or 90 | degree/order 70 |
| Initial state | 1 day | 1 day |
| Const. acceleration radial | 1 day | 1 day |
| Const. acceleration along-track | none | 1 day |
| Polynomial order 3 along-track | 1 day | none |
| Const. acceleration cross-track | 1 day | 1 day |
| Accelerometer scaling factors | 1 day, in all 3 directions | none |
| Pseudo-stochastic accelerations | 15 minutes, constrained: $1 \times 10^{-7} \frac{m}{s^2}$ | 15 minutes, constrained: $7 \times 10^{-6} \frac{m}{s^2}$ |

*2.4. Combination of Swarm and SLR on the Normal Equation Level*

To fill the gap between GRACE and GRACE-FO, we propose a combination of gravity fields derived by multi-SLR and Swarm analysis as detailed above. A very simple combination of SLR and GRACE, in fact the replacement of $C_{20}$ estimates in the monthly GRACE gravity fields by values derived from SLR, is already common practice [76]. In contrast to this, we perform a combination of Swarm and SLR on the NEQ level, i.e., we solve the equation

$$(w_{SLR}\boldsymbol{N}_{SLR} + w_{Swarm}\boldsymbol{N}_{Swarm})\boldsymbol{dx} = w_{SLR}\boldsymbol{b}_{SLR} + w_{Swarm}\boldsymbol{b}_{Swarm} \qquad (2)$$

for the vector of unknown gravity field coefficients $\boldsymbol{dx}$, where $\boldsymbol{N}_{SLR}$ and $\boldsymbol{N}_{Swarm}$ are the normal equation matrices of SLR and Swarm, $\boldsymbol{b}_{SLR}$ and $\boldsymbol{b}_{Swarm}$ are the corresponding right-hand side vectors of the individual normal equation systems, and $w_{SLR}$ and $w_{Swarm}$ are weighting factors. The solution of the combined normal equation system is superior to a combination on a solution level because correlations between gravity field coefficients and other force model, orbit, instrument and reference frame parameters are taken into account [34]. Since all satellite data have been processed consistently

and all but the gravity field parameters were pre-eliminated from the individual NEQs, the combination is straightforward.

The key question of the combination is the ratio of the weights $w_{Swarm}$ and $w_{SLR}$ assigned to the different observation techniques [38]. We first derive monthly weights by variance component estimation (VCE) [77]. As shown in Figure 1, the ratio $w_{Swarm} : w_{SLR}$ of the weights derived by VCE ranges from 20 to 95. To assess the plausibility of these weights, we derive alternative weights based on the accuracy estimates of the SLR ranges, which typically vary around 2 cm (see Table 1), and the GPS L1 phase observations of Swarm, for which Schreiter et al. [78] provide accuracy estimates close to 3 mm in case of strong ionosphere activity and close to 2 mm in case of low ionosphere activity (as long as no advanced observation screening in the region of the geomagnetic equator is performed). Based on these accuracy assumptions, a ratio of weights in the range from 44 to 100 can be expected, which is quite close to what is determined by VCE.

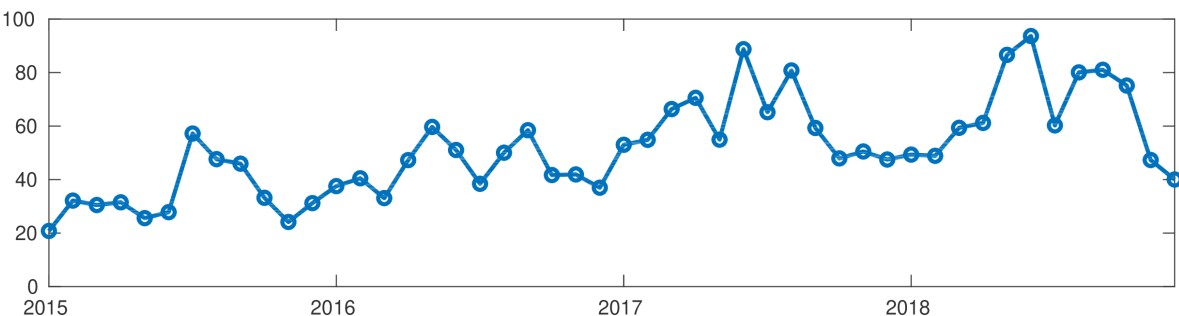

**Figure 1.** Ratios of monthly weights of the Swarm- with respect to the SLR-normal equations, as derived by variance component estimation.

The slightly positive trend in the relative weights visible in Figure 1 is explained by the decaying orbit altitude of the Swarm satellites and the corresponding increase in sensitivity to the gravity field. A seasonal variation probably is related to the seasonal tracking characteristics of the SLR stations and the fact that much more stations are located in the northern than in the southern hemisphere.

Test combinations were performed based on constant weighting ratios $w_{Swarm} : w_{SLR} = 100:1$, 10:1, 2.5:1 or 1:1. To assess the contribution of either Swarm or SLR to the combined solution, resolution matrices $\boldsymbol{R}_{SLR}$ and $\boldsymbol{R}_{Swarm}$ were determined from the individual normal equation matrices $\boldsymbol{N}_{SLR}$, $\boldsymbol{N}_{Swarm}$ and the inverse of the combined matrix $\boldsymbol{N} = w_{SLR}\boldsymbol{N}_{SLR} + w_{Swarm}\boldsymbol{N}_{Swarm}$ following the approach described in Sneeuw [79]:

$$\boldsymbol{R}_{SLR} = \boldsymbol{N}^{-1}\boldsymbol{N}_{SLR}, \tag{3}$$

$$\boldsymbol{R}_{Swarm} = \boldsymbol{N}^{-1}\boldsymbol{N}_{Swarm}. \tag{4}$$

The contribution numbers of the individual SHC are found on the main diagonals of the resolution matrices and are presented in triangle plots in Figure 2 for Swarm (left column) and SLR (right column). The contribution to the Sine-coefficients of the spherical harmonic spectrum are shown in the left hand part of the triangle plots, to the Cosine-coefficients in the right hand part, and the contribution to the zonal SHC (order 0) can be found in between. The contribution numbers vary between 0 (no influence) and 1 (determined by 100% from the corresponding observations). In the middle column of Figure 2, the mean contribution per degree is given.

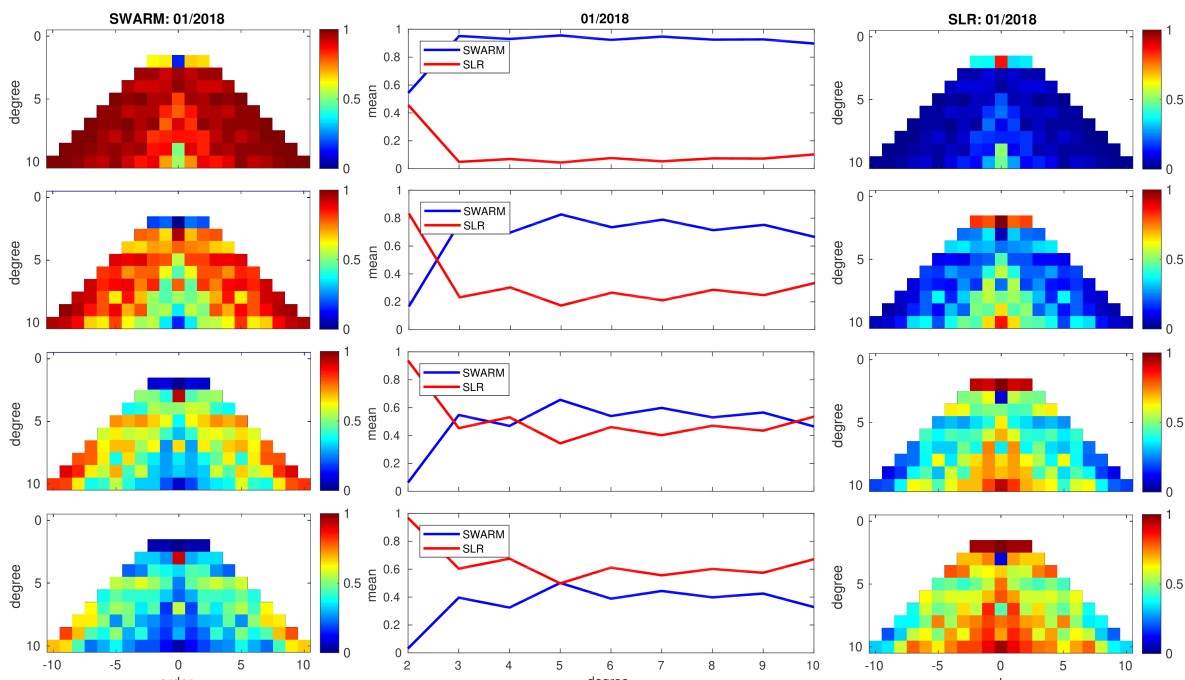

**Figure 2.** Contribution per spherical harmonics coefficient of Swarm (**left**) and SLR (**right**) satellites and mean contribution per degree (**middle column**) in case of relative weighting 100:1 (**top**), 10:1 (**second row**), 2.5:1 (**third row**) or equal weight (**bottom row**).

In case of a ratio of weights 100:1 (Figure 2, first row), only degree 2 SHC is significantly influenced by SLR, $C_{20}$ in fact is dominated by SLR. Applying relative weights of 10:1 (Figure 2, second row), still mainly degree 2 SHC are impacted by SLR. Considering the strength of the SLR-derived temporal gravity variations, the suppression of the contribution of SLR to the other SHC is not justified. A combination based on the weights derived by VCE (Figure 1) therefore is pointless.

Decreasing the relative weight of the GPS observations further (Figure 2, rows three and four), the contribution of Swarm is step by step reduced to the sectorial (equal degree and order) and near sectorial SHC, where the sensitivity of GPS observations is strong [70], and to $C_{30}$ that in case of SLR is weakly determined due to correlations with other zonal coefficients [68].

*2.5. Spectral Resolution, Signal Leakage, and Filter Loss*

The Earth gravity field is commonly represented by a spherical harmonic expansion (Equation (1)), truncated at a certain maximum degree $l_{max} < \infty$. The maximum degree (and consequently order) of this expansion determines the spatial scale of the represented signal. Monthly GRACE gravity fields are available to degree and order 90 (corresponding to a spatial scale of 460 km at the equator), Swarm monthly gravity fields are expanded to degree/order 70 but contain significant time-variable signal only up to about degree 13 (approx. 3100 km at the equator) [47]. SLR-derived gravity fields are even limited to degree 6 (approx. 6700 km at the equator) due to correlations between individual SHC at higher degrees that cannot be determined in unconstrained solutions given the inhomogenous observation coverage [10].

The truncation of the spherical harmonic expansion at a certain maximum degree causes signal leakage (e.g., [23–26]). To demonstrate this effect, we simulate a mass layer with uniform mass distribution within the island of Greenland, while the mass over the rest of the globe is set to zero. The simulated mass distribution is expanded in a series of spherical harmonics and reconstructed from the SHC truncating the series at various values of $l_{max}$. In Figure 3a, the reconstructed mass distributions are shown and the percentage of the integrated mass still contained inside the shorelines of Greenland is listed.

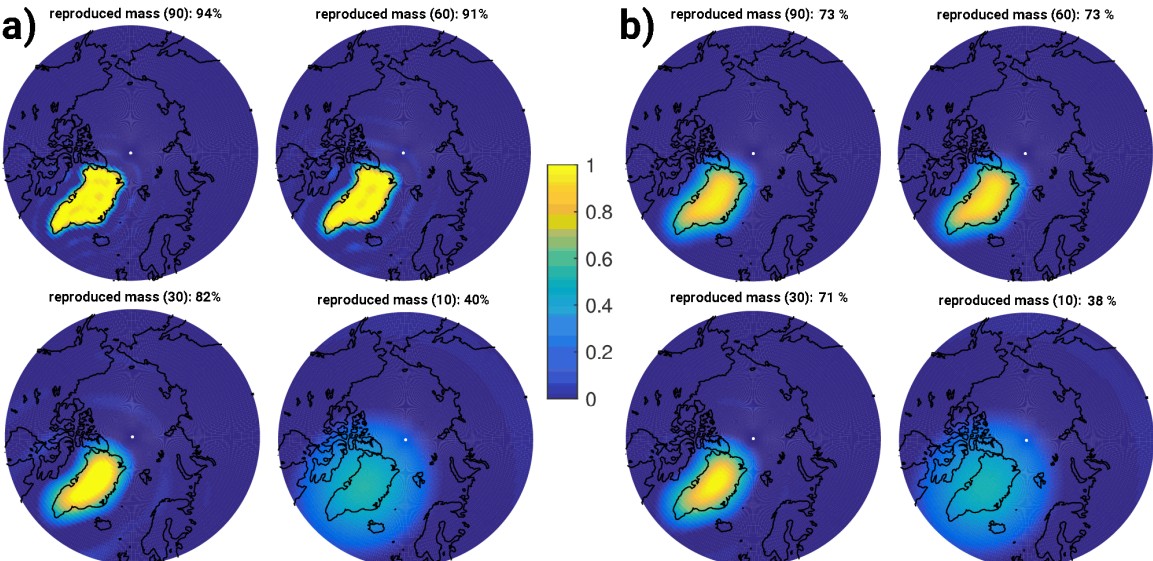

**Figure 3.** Integrated mass within Greenland reconstructed at different truncation degrees, (**a**) without filter or (**b**) smoothed by a 300 km Gaussian filter.

The signal content is also attenuated by filters to suppress the noise inherent to the gravity field models. A very simple filter that is commonly applied to GRACE monthly gravity fields is a Gaussian filter with 300 km filter radius [27]. In the spatial domain, the Gaussian filter represents a weighted average, the weighting function being bell-shaped with 300 km the half-width radius. In the spectral domain, the filter corresponds to degree-dependent scaling factors that quickly approach zero for higher degrees (from 1 at degree 0, the scaling factor has already dropped to 0.01 at degree 60). The experiment on the signal leakage by truncation of the spherical harmonic expansion is repeated, additionally applying a 300 km Gaussian filter for smoothing (Figure 3b). Due to the filter attenuation at medium to high degrees, the signal loss is even more dramatic than in the unfiltered case. More advanced filter types like the non-isotropic filter proposed by Han et al. [28] and the decorrelation filter DDK [29] were designed to minimize the filter loss, but the principle problem of signal attenuation persists.

Both experiments show that gravity fields truncated at different maximum degrees cannot be compared. At the very limited resolution of the SLR-derived gravity fields, only a fraction of the original mass is localized inside the borders of Greenland. The true mass localization and change can be recovered approximately applying iterative forward-modeling approaches [25,26], which require additional assumptions like mass loss being concentrated in fast flowing sections of ice streams, or at the coast. Nevertheless, leakage (in and out) and filter loss remain major limitations to the quantification of mass transport from satellite gravimetry. In the following, we truncate all gravity fields to the same low degree for the sake of comparison. Furthermore, we refrain from the use of filters.

To derive mass from SHC, first the dimensionless SHC are scaled by the factors provided by Wahr et al. [30] to transform them to EWH. Then, global 1°-grids of EWH are computed from the scaled SHC by spherical harmonic synthesis (Equation (1)). The series expansion is truncated at the maximum degree of choice (either 6 or 10). To transform the EWH grids to mass, each grid cell is multiplied by its area (dependent on latitude) and the density of water (1000 kg/m$^3$). By integration over the area of interest, e.g., Greenland, the final mass estimates are derived.

## 3. Results and Discussion

Figure 4 shows the integrated secular effect of glacial isostatic adjustment (GIA), ice mass and snow mass change in the polar regions during the period 2010 to 2014 as derived from un-smoothed degree 60 monthly gravity field models determined from GRACE GPS and K-band observations.

The mass loss is mainly related to ice loss (dynamic or by melting and run-off). GIA, i.e., the relaxation of the crust in reaction to the large scale ice melt after the last ice age, counteracts ice mass loss and therefore the actual ice mass loss is even larger (estimates for the mass change induced by GIA vary from 1 Gt/year to 20 Gt/year for Greenland and from 55 Gt/year to 110 Gt/year for Antarctica [32]) than indicated by the figures, while snow mass depends on the season and largely cancels out in a multi-year mean. The observation coverage is densest near the poles due to the polar orbits of the GRACE satellites and consequently the noise is lowest near the poles. At lower latitudes, the noisy striping typical for GRACE becomes visible.

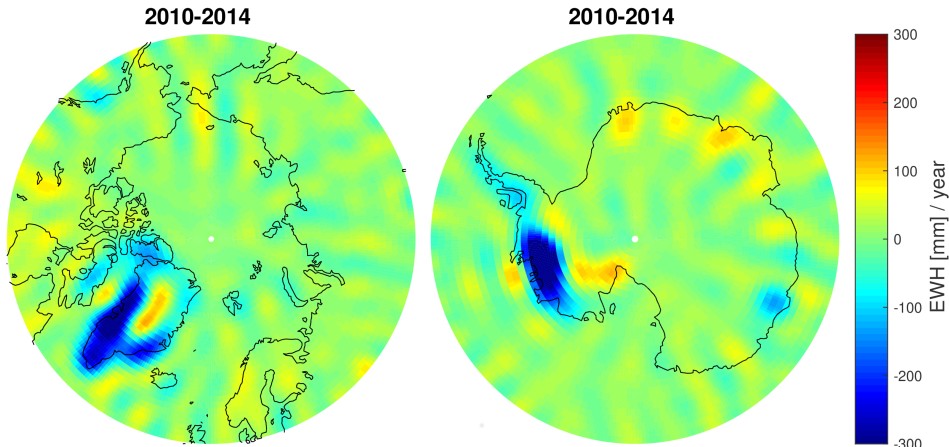

**Figure 4.** Trends in equivalent water height (EWH) as observed by GRACE in the Arctic (**left**) or Antarctic (**right**) region in the period 2010-2014, derived from unfiltered d/o 60 solutions.

For comparison to SLR-derived gravity field models the GRACE results are truncated at degree/order 10 and the corresponding EWH trends are shown in Figure 5. The trend signal that is well localized over the continents in Figure 4 leaks out over the oceans in Figure 5, as predicted by our simulation (Figure 3).

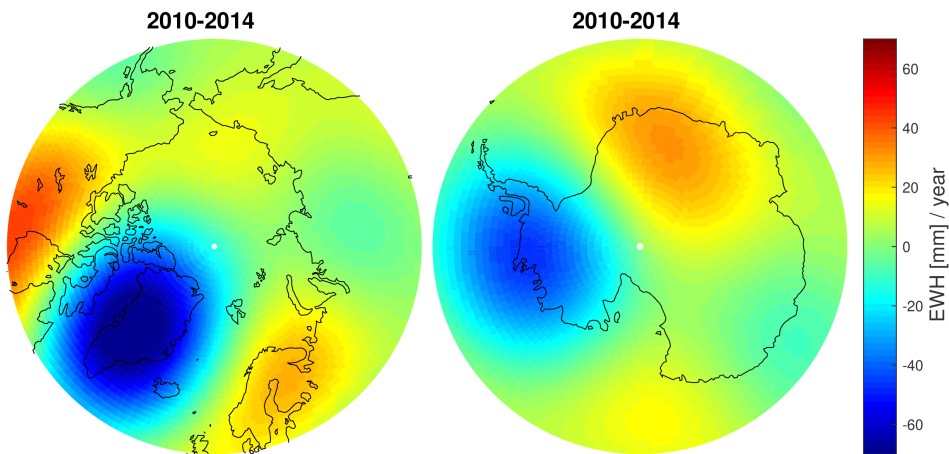

**Figure 5.** Trends in equivalent water height (EWH) in polar regions derived from GRACE gravity fields truncated at d/o 10.

In Figure 6, global plots of GRACE-derived trends 2010–2014 in EWH (left) and the amplitudes of seasonal variations (right) are provided, both truncated at degree/order 10. Trends are mainly visible at high latitudes, where large scale ice mass loss is the main reason for negative trends, GIA for positive trends in North America and Fennoscandia. At this low spherical harmonic resolution, it is difficult to state if negative trends near the west coast of North America are related to drought. Positive trends in the Amazon region probably are related to inter-annual variability that is not strictly seasonal.

Seasonal variations (Figure 6, right) are related to hydrology and are strongest in the tropical and subtropical regions with strong seasonal variation in rainfall.

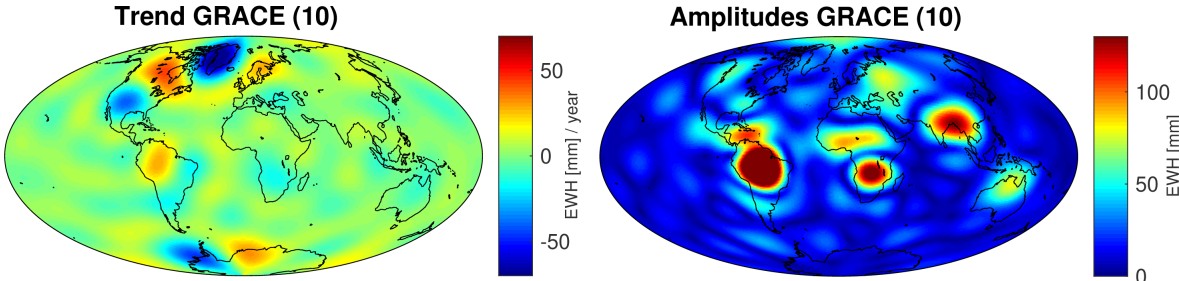

**Figure 6.** Global trends 2010–2014 in EWH (**left**) and amplitude of annual EWH variations (**right**) derived from GRACE gravity fields truncated at d/o 10.

Figure 7 provides the same information as Figure 6, but derived from SLR. In the top row, the unconstrained degree 6 gravity field models, and, in the bottom row, the constrained degree/order 10 solutions are evaluated. Especially in the case of the EWH trends, the improvement of the localization with a higher maximum degree is obvious. In the case of seasonal variations, the amplitudes in all but the Amazon basin are heavily attenuated compared to GRACE (Figure 6, right).

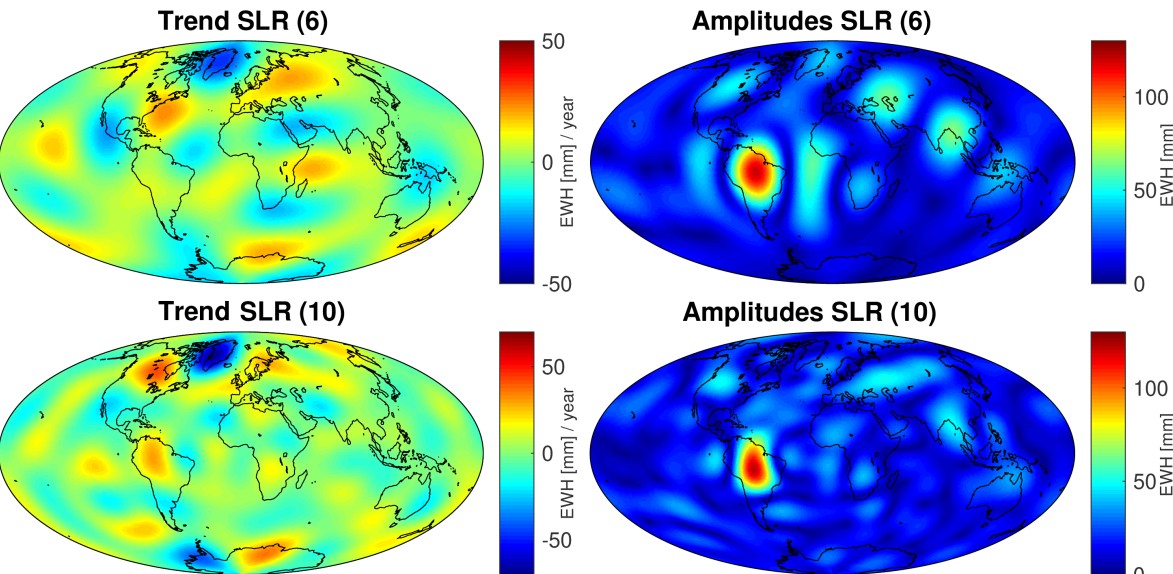

**Figure 7.** Global mass trends (in equivalent water height) as observed by SLR between 2010 and 2014 (**left**) and amplitude of annual variations (**right**). Top: unconstrained monthly gravity fields up to degree 6, bottom: constrained monthly gravity fields up to degree 10.

Figures 8 and 9 focus on EWH trends derived from SLR for the time periods 1995–1999, 2000–2004, 2005–2009 and 2010–2014 in polar regions. The trends are determined either from unconstrained degree 6 solutions (top row) or from regularized monthly gravity field models up to spherical harmonic degree and order 10 (bottom row). While we can confirm the findings of Bonin et al. [33] based on simulations that with degree 5 gravity fields, the correct localization of mass variations in Greenland and Antarctica is not granted; we observe rather precise localization of the SLR-derived mass trends in Greenland or Antarctica for the regularized degree 10 solutions (compare the sub-figures for the time-period 2010–2014 in Figures 8 and 9 with the corresponding results from GRACE in Figure 5). Along the coasts of Greenland and close to the coast of West Antarctica, the SLR gravity fields indicate significant mass loss over the ocean, but this has to be expected taking the effect of leakage into account (Figure 3).

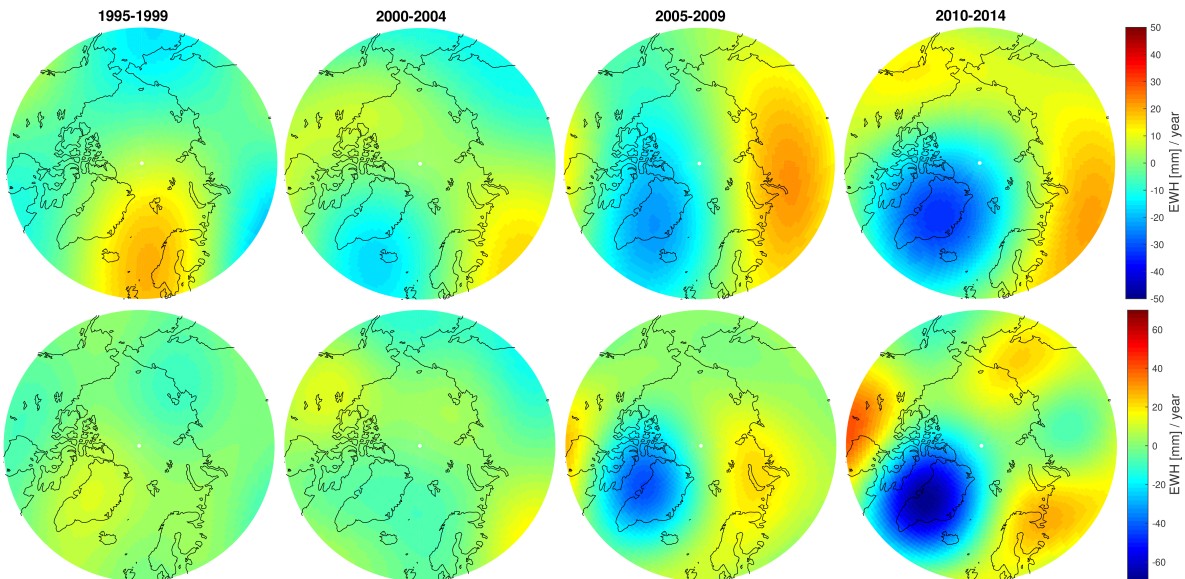

**Figure 8.** Trends in EWH as observed by SLR in the Arctic region during 5-year periods between 1995 and 2014. **Top**: unconstrained monthly gravity fields up to degree 6, **bottom**: constrained monthly gravity fields up to degree 10.

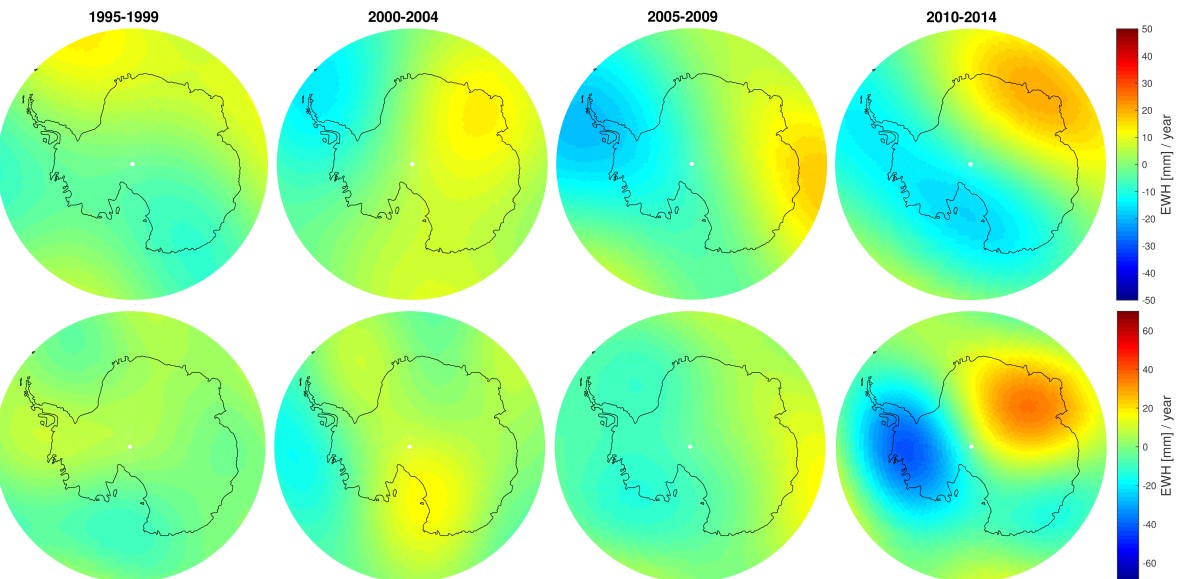

**Figure 9.** Trends in EWH as observed by SLR in the Antarctic region during 5-year periods between 1995 and 2014. **Top**: unconstrained monthly gravity fields up to degree 6, **bottom**: constrained monthly gravity fields up to degree 10.

In the following, time-series of mass change within certain regions are studied to demonstrate the accordance of SLR and Swarm with GRACE (both truncated at degree/order 6 corresponding to the unconstrained SLR solutions). SHC of degree 1 and $C_{20}$ are removed, the latter because the GRACE estimates of $C_{20}$ suffer from temporal aliasing [13] and accelerometer instrument noise [80] (the effect of $C_{20}$ alone on mass change in Greenland sums up to about 100 Gt in the time period from 2000 to 2018). No re-scaling to compensate for signal leakage is applied and no corrections for GIA and seasonal variations due to snow mass are applied. For a complete treatment of ice mass loss as derived from GRACE data, we refer to the literature in this field (e.g., [4]).

While GRACE observations cover the period 2002–2017, earlier estimates of mass change have to be based on observations of the geodetic SLR satellites. After the end of the GRACE mission,

mass estimates are based either on SLR or on Swarm observations. The good agreement of SLR- and GRACE-derived seasonal variations in the Amazon basin is obvious in Figure 10, while Swarm tends to slightly overestimate the seasonal variation.

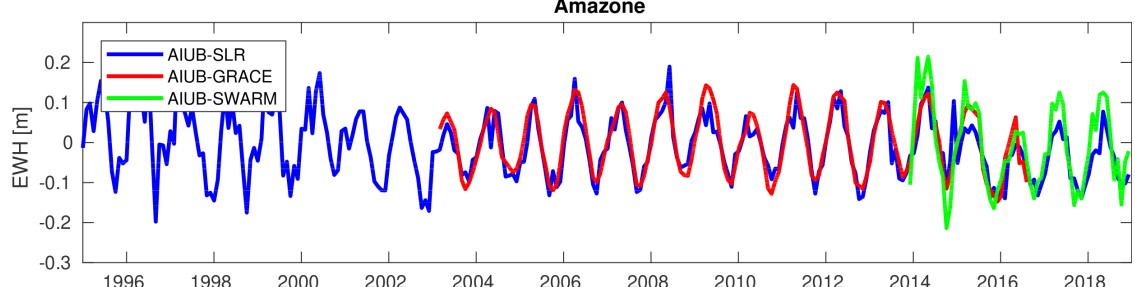

**Figure 10.** Monthly mass estimates for the Amazon basin. GRACE and Swarm gravity fields were truncated at degree 6 to match the resolution of SLR gravity fields. Spherical harmonic coefficient $C_{20}$ was excluded from the comparison.

The evolution of the Greenland mass is shown in Figure 11. Again, the long-term agreement of all three observation types is very good. SLR does not reveal significant mass loss in Greenland in the pre-GRACE era. Obviously, the GRACE mission was launched just in time to observe the onset of ice loss in Greenland. After acceleration in the ice mass loss, the GRACE mass estimates almost level out in the time period from 2014 to 2016. SLR tends to slightly overestimate the Greenland mass loss between 2014 and 2017, while, after 2017, even a small mass gain is visible in both the SLR and Swarm derived mass estimates. The reduction of the SLR- and Swarm-derived mass loss therefore is time-shifted by about three years compared to GRACE. Bonin et al. [33] also observe diverging mass trends for Greenland but state that, compared to the independent time-series of mass change derived by the Ice-sheet Mass Balance Inter-comparison Exercise (IMBIE) for Greenland and Antarctica [81], in the long term, these deviations of SLR average out. The inter-annual variability is larger in the monthly SLR estimates than observed by GRACE at this low spherical harmonic degree, and even larger in the monthly Swarm solutions.

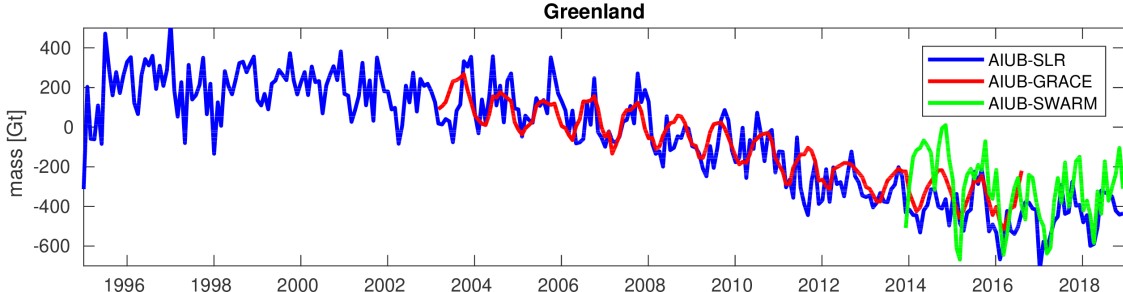

**Figure 11.** Monthly mass estimates for entire Greenland (see masks in Figure 6). GRACE and Swarm gravity fields were truncated at degree 6 to match the resolution of SLR gravity fields. Spherical harmonic coefficient $C_{20}$ was excluded from the comparison.

While the accuracy of the GRACE mass estimates at this very low resolution can considered to be constant in the time-period shown, because temporal variations in noise due to the changing satellite environment and technical problems mainly are manifest in the weakly determined high degree/order SHC, the quality of the SLR-derived estimates depends on the station network that was even more sparse in the 1990s than it is today, and on the number of satellites contributing to the monthly solutions. A reduction in the scatter can be observed after the launch of LARES in 2012. The monthly Swarm solutions exhibit even greater scatter than the SLR solutions and additionally seem to over-estimate the seasonal variation compared to GRACE. After 2014, the Swarm and the SLR results match well in trend

and in phase. During 2014, the Swarm results are impaired by high ionosphere activity, non-optimal GPS receiver settings, and in the first half of 2014 by reduced observation sampling [75].

Consecutively, mass trends were determined for the coastal regions of Greenland with strong mass loss, the region of the inland ice sheet, where a weak gain in mass is observed by GRACE, and the coast of West Antarctica. The masks used are provided in Figure 12. The separation of Greenland into regions of mass gain and regions of mass loss was done based on Figure 4. The mask of Antarctica and its glacial sub-basins is derived from Horwath and Dietrich [82].

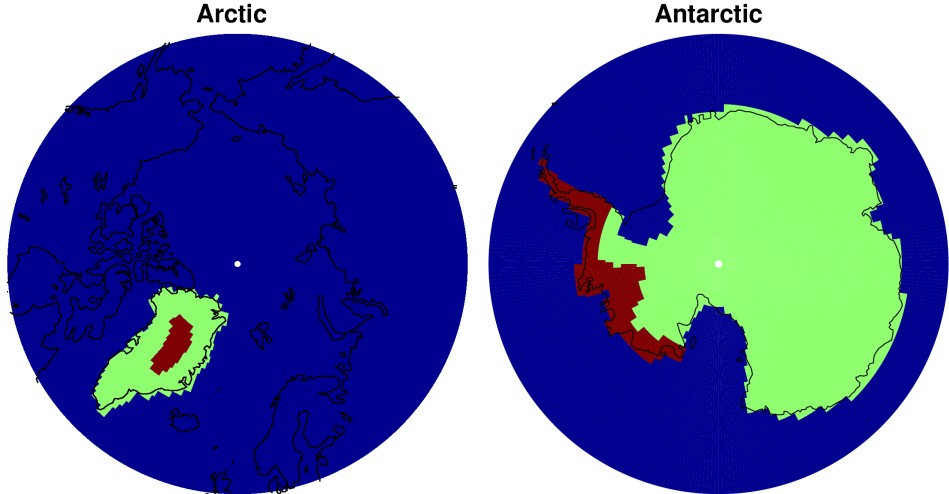

**Figure 12.** Masks defining inland and coastal regions of Greenland (**left**) and the coast of West Antarctica (**right**). The resolution of the masks is 1°.

Starting with the first monthly estimates from SLR in 1995, we fit deterministic models including bias, trend and seasonal variations within 5-year periods to integrated mass estimates of either SLR or GRACE monthly gravity field models summed up over the coastal areas (Figure 13, left) or the inland ice sheet (Figure 13, right) of Greenland, or the coast of West Antarctica (Figure 14).

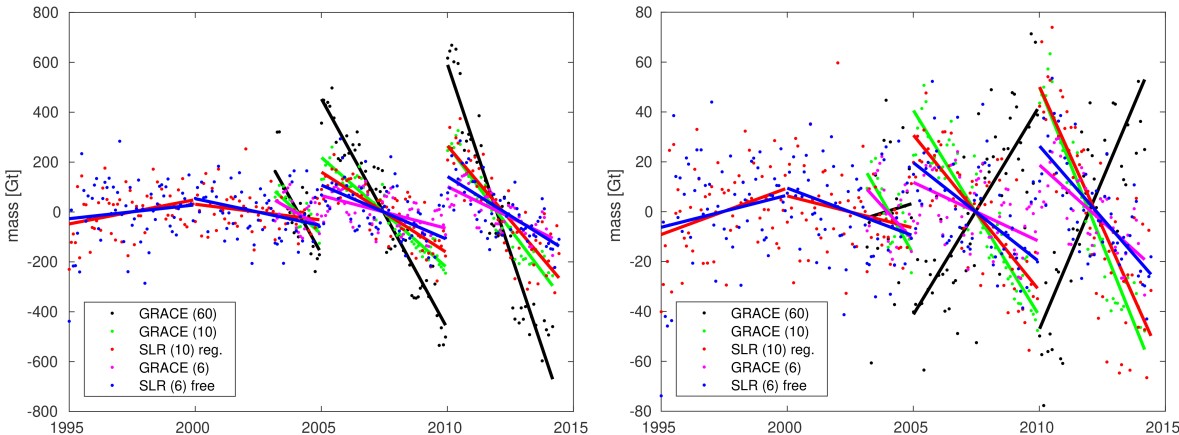

**Figure 13.** Monthly mass estimates in Greenland from GRACE and SLR, and best fitting trends for 5-year periods for the region of mass loss along the coast (**left**) or the inland region of mass gain (**right**). Each 5-year period is centered around zero.

In case of SLR, the trend estimation is based either on the unconstrained degree 6 monthly gravity field solutions or on the regularized degree 10 solutions (compare Figures 8 and 9 for the corresponding spatial plots). For comparison, the original degree 60 GRACE gravity fields were truncated at degrees 6 or 10 and also mass trends in 5-year intervals determined.

The SLR trend estimates reveal good agreement with GRACE (truncated at the corresponding degree) for the time periods where a direct comparison is possible. However, due to signal leakage at the very low resolution of SLR, the absolute mass loss is drastically underestimated. Truncated at degree 6 or 10, due to the very limited spatial resolution, neither GRACE nor SLR are able to trace the inland mass gain observed by GRACE at higher resolution (Figure 13, right).

Prior to the start of GRACE, no distinct mass trend within Greenland can be observed. Since the start of the highly sensitive GRACE observations, the mass loss at the coast is accelerating within the time period covered by Figure 13. This fact has already been reported, e.g., by Velicogna [83] and recently by Bevis et al. [84], and is visible in SLR and GRACE estimates alike. In case of the degree 6 SLR solutions, the five year trend estimates are 2000–2004: −21.5 Gt/year, 2005–2009: −43.6 Gt/year, 2010–2014: −63.1 Gt/year. As explained above, due to leakage, these values are drastically underestimated compared to the degree 60 GRACE solutions 2003–2004: −179.8 Gt/year, 2005–2009: −184.4 Gt/year, 2010–2014: −302.7 Gt/year (no GIA corrections applied).

Comparable results are achieved for the area of significant ice mass loss along the coast of West Antarctica (Figure 14), where, due to the bad localization of the mass loss in the degree 6 solutions, the five year trends derived from the degree 10 SLR gravity fields 2000–2004: −5.4 Gt/year, 2005–2009: −15.1 Gt/year, and 2010–2014: −50.7 Gt/year are quoted here together with the degree 60 GRACE solutions 2003–2004: −85.5 Gt/year, 2005–2009: −121.1 Gt/year, and 2010–2014: −201.0 Gt/year. Again, the trend estimates at the given low degree of SLR are underestimates due to leakage, but the increase in ice melt is captured very impressively.

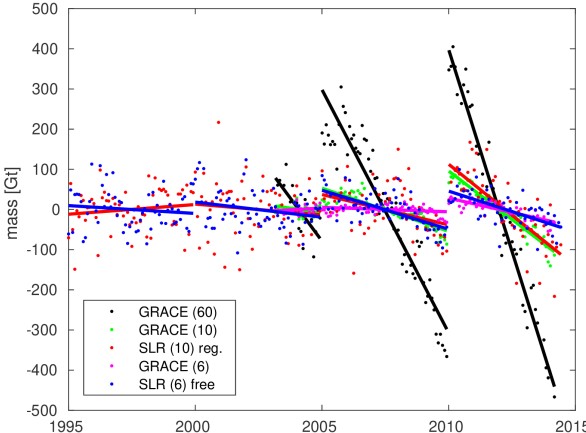

**Figure 14.** Monthly mass estimates at the coast of West Antarctica from GRACE and SLR, and best fitting trends for 5-year periods. Each 5-year period is centered around zero.

The individual and the combined monthly mass estimates within Greenland for the two extreme cases of 100:1 and 1:1 relative weighting of Swarm and SLR are shown in Figure 15. The combination starts with January 2015 because, in 2014, the GRACE data are still quite complete while the Swarm data is impaired by strong ionosphere activity and non-optimal GPS receiver settings (receiver settings were adapted in May 2015). During 2015 and in the first half of 2016, a direct comparison to GRACE results is possible. After August 2016, the accelerometer on GRACE B was completely switched off and the processing of GRACE data at AIUB stopped. From the direct comparison, the impression that Swarm generally slightly overestimates the amplitude of the mass variations while SLR overestimates the secular mass loss is confirmed.

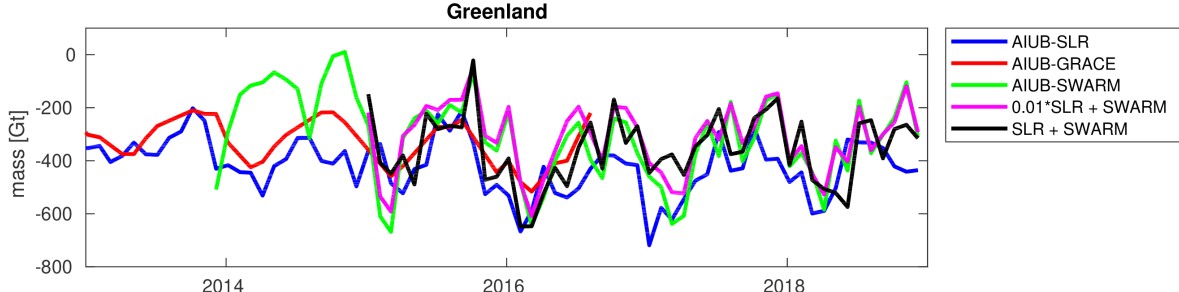

**Figure 15.** Monthly mass estimates within Greenland. All time-series truncated at degree 6 and with $C_{20}$ excluded.

In case of relative weighting 100:1, the combined mass estimates closely follow the Swarm-only results. This can be expected in view of the results of the contribution analysis (Figure 2), even more so because $C_{20}$ is excluded from the analysis. In the case of equal weights, the combined solution follows more closely the GRACE mass estimates, but still with higher inter-annual variability.

The monthly differences between GRACE and the individual or combined alternative mass estimates are shown in Figure 16. During 2015, the combination of SLR and Swarm with equal weighting has the smallest differences with respect to GRACE; in the first half of 2016, the combination 0.4*SLR+Swarm is slightly closer. The RMS of the differences with respect to GRACE over all months is 166 Gt for the SLR-only solutions, 110 Gt in case of Swarm-only, 110 Gt for the combination 0.01*SLR+Swarm, 83 Gt for 0.1*SLR+Swarm, 67 Gt for 0.4*SLR+Swarm and 68 Gt in case of equal weighting.

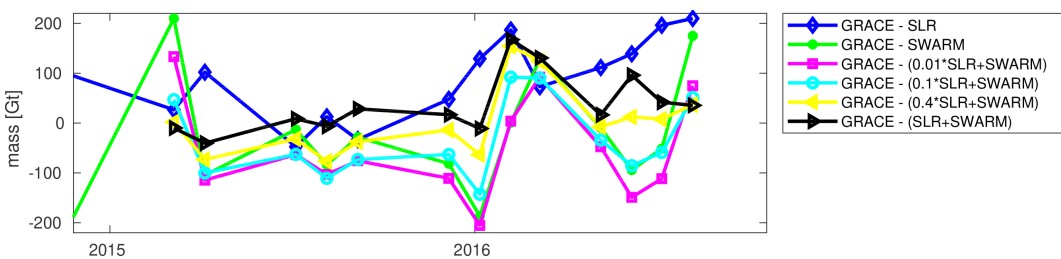

**Figure 16.** Monthly differences between GRACE and SLR (blue), Swarm (green) or combined mass estimates within Greenland. Missing values indicate gaps in the GRACE time-series.

These results clearly indicate that a combination with close to equal weights of Swarm (GPS) and SLR is closer to the GRACE reference than when weights are based on variance analysis of observation residuals. This can be explained by systematic errors in the kinematic orbits due to the GPS phase processing. In fact, GPS also has to be down-weighted in combination with the GRACE K-band observations (by a factor of approx. 200, see [34]). Recent experiments using one month of kinematic GRACE orbits derived with undifferenced integer-fixed GPS phase ambiguities indicate that the inconsistency between K-band and GPS is reduced. A better assessment will be possible as soon as longer time-series of kinematic orbits based on undifferenced integer-fixed ambiguities become available, but this will be the topic of a future publication.

## 4. Conclusions

Satellite laser ranging can contribute to the derivation of mass change estimates at the spatial scale of the Amazon basin, Greenland, or West Antarctica prior to the GRACE mission. SLR and high-low tracking of the Swarm satellites both may be used to bridge the gap between GRACE and GRACE-FO concerning the large scale mass loss in the polar regions of the Earth. When all individual gravity field results are truncated at the same spherical harmonic degree, the results can be compared directly. They match well for Greenland with a somewhat higher inter-annual variability of the SLR monthly mass estimates. Swarm can only be included in the comparison near the end of the life-time of the

GRACE satellites. The seasonal variations visible in Swarm monthly mass estimates seem to be slightly over-estimated. On the other hand, the SLR derived mass loss within Greenland is larger than what can be extracted from GRACE observations between 2014 and 2017. After 2017, both SLR and Swarm results indicate moderate mass gain within Greenland.

For a correct localization of mass change signal along the coast of West Antarctica, a spherical harmonic expansion up to degree 10 is desirable. This can be achieved by the regularization of SLR-only gravity field models or, preferably, by a combination with other LEO, e.g., Swarm data. A combination on the normal equation level, correctly taking into account all correlations between estimated and pre-eliminated parameters, is feasible as long as all satellite data is processed consistently. The best fit between a SLR/Swarm combination and GRACE for the period from January 2015 to August 2016, where both Swarm and GRACE observations of good quality are available, is achieved when SLR and Swarm normal equations are combined with almost equal weights.

It is foreseen to continue the combination of SLR data, taking into account the contributions by different analysis centers, and the combination of SLR with Swarm and possibly also GRACE-FO on the normal equation level in the frame of the Combination Service for Time-variable Gravity field models (COST-G) [34,52], the newly established product center of the International Gravity Field Service (IGFS) under the umbrella of the International Association of Geodesy (IAG).

**Author Contributions:** GRACE processing, derivation of mass trends, conceptualization and writing—U.M.; SLR processing and investigation—K.S.; determination of kinematic GRACE and Swarm orbits—D.A.; Swarm processing—C.D.; software development, supervision and funding—D.T.; supervision and project administration (SLR)—R.D.; supervision and project administration (GRACE, Swarm)—A.J.

**Funding:** This research was partly funded by BKG "Vertrag über die Weiterentwicklung der Berner GNSS Software, 21.04.2017" and by the ESA project Swarm-DISC, contract No. 4000109587/13/I-NB.

**Acknowledgments:** We want to thank three anonymous reviewers for their invaluable comments.

**Conflicts of Interest:** The authors declare no conflict of interest. The funders had no role in the design of the study; in the collection, analyses, or interpretation of data; in the writing of the manuscript, or in the decision to publish the results.

## Abbreviations

The following abbreviations are used in this manuscript:

| | |
|---|---|
| AC | Analysis Centre |
| AOD | Atmosphere and Ocean De-Aliasing |
| CMA | Celestial Mechanics Approach |
| GIA | Glacial Isostatic Adjustment |
| GPS | Global Positioning System |
| Gt | Gigatons ($10^{12}$ kg) |
| LEO | Low Earth Orbiter |
| NEQ | Normal Equation |
| PPP | Precise Point Positioning |
| RMS | Root Mean Square |
| SHC | Spherical Harmonic Coefficients |
| SLR | Satellite Laser Ranging |
| VCE | Variance Component Estimation |

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
