# Peer review of "SLR, GRACE and Swarm Gravity Field Determination and Combination"

_remotesensing, doi:10.3390/rs11080956_

Round 1

Reviewer 1 Report

pre.cjk { font-family: "Nimbus Mono L",monospace; }p { margin-bottom: 0.1in; line-height: 120%; }Dear authors, It was a pleasure to read your manuscript. In my opinion it contains very interesting information, sufficient for publication in  Remote Sensing. I have indicated however that it is also my opinion that a major revision is required. This revision primarily concerns textual issues, a need for more explanation and details for certain parts, and a bit of restructuring to improve the flow of  your manuscript. Concerning the restructuring, please see my comments related to Figs. 1 and 2. Concerning a need for more details, see  e.g. the annotation at Table 1 and especially my annotations related to regularization/stabilization of gravity field solutions (important to know for e.g. reproducibility). I have thus provided an annotated version of your manuscript.  I think/hope most of my annotations speak for themselves. I am looking forward to reading the revised version of your manuscript. Kind regards, Reviewer

Author Response

Please see the attached PDF of the rebuttal.

Reviewer 2 Report

Review of “SLR, GRACE and SWARM gravity field determination and combination” by Meyer et al.

The authors describe the generation of gravity field solutions based on three different types of satellite observations, as given in the title. They compare the respective solutions and the related mass signals over the Greenland Ice Sheet (GrIS). Finally, they combine SLR and SWARM observations on the normal equation level in order to find an optimal fill-in for the GRACE/GRACE-FO gap. This investigation of an optimal SLR/SWARM combination is again carried out by evaluating mass changes in Greenland.

The paper adds value to the research in the field of satellite gravimetry and, consequently, geophysics, where the community utilizes satellite gravimetry, particularly by the final part, the SLR/SWARM combination. In this sense, I think, the paper deserves to be published in Remote Sensing. However, in its current shape the writing is somewhat erratic, the language is sometimes poor, and the methods and results are not transported very clearly, making it fairly unaccessible to readers without the same background as the authors. Therefore, I recommend to request major revisions from the authors before acceptance. I have listed a few suggestions below.

My major points of criticism are:

1) The paper lacks a clear outline and aim.
Not much of the findings, such as GrIS mass loss, or its timing, or the effect of leakage is novel. However, as the paper addresses these things over long passages, it is easily forgotten what novelty the authors do actually present to the community in their paper.  Instead, the whole methodological innovation is introduced only in section 5, near the end of the paper. I suggest to stress more clearly in the introduction and in the main text what is novel and what is just done to show that the results derived using single methods (i.e. section 3 and 4) are sensible.
I think a clearer classic distinction between data/methods, results, discussion, conclusions should be adopted to help the reader understand what was done. (Note that section 6, “discussion”, is actually a summary/conclusion.)
Also, because section 3 on filtering and leakage, including the respective conclusions that no filtering is carried out and that leakage poses a big problem at the achievable low degrees, does not contain novel findings and does not even discuss the actual satellite observations, it could be dealt with references to qualified scientific literature only, as part of a newly set-up methods section.

Coming back to the novel achievements, it is certainly interesting that equal (or close) weights for SWARM and SLR are so much more favorable if the nominal accuracies would require factor 100. I think the paper would gain a lot of strength if it was able to come up with some possible answers to this conundrum. Is it that the SWARM accuracy is overly ambitious? Would a better measure for accuracy possibly be the (longterm-average?) difference between SWARM and GRACE and SLR and GRACE, respectively?

2) Selective application:
The data are only ever tested over the Greenland Ice Sheet. There are figures of Antarctic mass trends, too, where the different satellite data are introduced, but they are rather unmotivated and not much discussed. However, satellite gravimetry offers a much wider range of applications, in hydrology, oceanography, and seismology. (Also, in the very first sentence of the abstract, “ice mass change” is mentioned as if it was not part of “large scale mass transport in the system Earth”, which it actually is.) I think the manuscript would massively benefit from a broadening of the scope. Why not for example show global trends, even if at low degrees only? Or, as the number of spherical harmonic coefficients that are actually evaluated in most of the analysis is quite low, why not show per-coefficient monthly solutions? Then, depending on the shown quantities, other applications such as large-scale hydrology could be highlighted, too (at least briefly).

3) Details of satellites, satellite data, processing
The description of data and methods should definitely be improved. This could start with including GRACE and SWARM satellites in a re-defined table 1, where the differences in the respective satellite sources and techniques would be more obvious.
Then, as the whole analysis is naturally based on spherical harmonics, there should at least one equation that gives a spatial field as its spherical harmonic expansion (possibly early on, as spherical harmonics and degrees are mentioned early in the introduction). Otherwise, only spherical-harmonics literates are able to interpret mentions of “degree”, “order”, “C_20” (many instances) or the “Cos-/Sin-terms“ (L319/320).
What irritates me most is the unmotivated description of processing details in tables 2-4 (background force model, “parameterization” whatever that exactly means). The table captions do not help to explain the content of the tables. Take, for example, table 3, “station coordinates” are given as “30 days”, this does not make sense to non-experts (or maybe not even to experts). Is it an update/observation frequency? Next item is “Earth orientation parameters”, the entry says “piecewise linear daily” which is again obscure. Many terms are also not understandable to non-experts, such as osculating elements, “once-per-rev”, etc. It might be best to drop all three tables and include respective information in the main text or in an appendix, making sure that things are properly explained or, where this would lead too far, simply summarized and referenced properly.
Another issue is that certain terms regarding the satellites are used synonymously, which sometimes obscures the meaning to the reader. For example, LEO seems to be used as a synonym for SWARM sometimes, and sometimes not. K-Band observations is used instead of GRACE observations. I suggest to harmonize the respective terminology throughout the manuscript.

4) Lack of references to scientific literature
As mentioned before, many of the findings are not exactly new. However, the authors tend to cite selectively. I give three prominent examples here:
- Leakage: L44-46 and L211 onwards; the only referenced study is Chen et al 2015, which discusses leakage in West Antarctica, with a very specific approach to tackle it. Almost any scientific article utilizing GRACE observations for a specific geophysical problem will have discussed how leakage was treated, so the literature acknowledged in this context should be broader.
- Onset (and stagnation) of Greenland ice loss: L254-256. This has been reported in other studies, e.g. Sasgen et al. (2012), Bevis et al. (2019).
- The gap between GRACE and GRACE-FO has been addressed by many studies. Examples are Rietbroek et al. (2014), Peralta-Ferriz et al. (2016), an armada of conference abstracts from EGU and AGU meetings of the past years, and even some articles that the authors cite in different contexts, such as [65].

5) Language
The language is often jargon, i.e. inaccessible to non-experts. The authors should make an effort to make their paper more comprehensible. Some instances will be among the list of smaller comments below. Others are
- L78: “biased sum”
- L95: “pseudo-observations”
- L101: “high dynamical stiffness”
- L105: “long-periodic systematics”
- L170/175: “absorb (... variations)”
- L178 “high sensitivity of the K-band observable”
Another issue is that often logical connections between two sentences are not clear, for example (including possible logical connections in caps):
- L98-100: “IN THESE CASES, all information about gravity variations is derived from the orbit dynamics”.
- L114: “Therefore most of the geodetic SLR satellites have a low area to mass ratio, THUS REDUCING THE SURFACE FORCES ACTING ON THE SATELLITE”.
- L276: “In the latter case”; “latter” probably refers to the degree 10 solutions, but the last instance before its mention is the reference to Figures 1/2 in brackets.
Also, I believe that the English needs revision, specifically the use of hyphens.

I append a list of smaller/technical comments:

Abstract: Some numbers which underpin the findings stated in the abstract should be inserted.

The references are apparently in alphabetical order, but should be in the order of appearance in the text, e.g. [56] in L15 should be [1]. Also, references included in the sentence structure should be written out, e.g. L29 “Cox and Chao [13] [...] focus on variations...”.

L17 and elsewhere: “But” --> “However,”?

L25 and elsewhere: “time-variation” “time-resolution”, consider using “temporal” instead.

L25-L26: If the added value of LARES (temporal variation at lowest degrees) is due to the lower orbit, please state this more clearly.

L33: Why are SWARM satellites well suited to bridge the gap?

L43 “the derived gravity fields”

L49 and L250, state at which degree the spectra were truncated.

L53 and elsewhere: This should be “coast of West Antarctica”, because “West Antarctica” is a standing, well defined name for a geographic region where otherwise East and West are not necessarily good absolute categories (in contrast to the US West coast, for example). Also, it should be noted in this context that mass variations in Greenland and West Antarctica are mostly related to ice mass loss.

L60 (and L70): The mention of GOCE is a bit unmotivated here. Perhaps, a more general satellite gravimetry section could introduce GOCE less abruptly.

L61-65: The last two sentences are somewhat obscure. For example, is the “combination approach” that the authors follow, a defined recipe? If yes, this should be clearly stated and referenced. In accordance with my major comment 1, I suggest to rewrite the last part of the introduction so that the outline for the whole paper is clearer.

L74-81: A few more references are needed here.

L75: Doris-tracking; describe what it is and why it is ignored.

Table 1: “error” in caption, “sigma” in table. Harmonize?

Table 2: Why is the d/o for GRACE and SWARM so much higher than the actual solutions produced (e.g. 60-90 for GRACE)?
Apparently, for each row that has “d/o” in the “Resolution” column, the right columns give spatial resolution in terms of degree and order, and unreferenced/binary values else. This is really not intuitive and should be changed. (see also my major point 3 above).

L130: Explain briefly what GIA is. “Snow cover” should be “snow mass”, as snow cover is often understood as binary: Snow or no snow at a pixel/location.

L141: Why keep the SHC at deg=6, order=1?

Figure 1/2: It is not clear in this context how the mass trend has been obtained exactly, and the respective description appears only much later in the text (section 4). In accordance with my above suggestion to introduce a clearer structure, I suggest to move these figures to what could be a real results section.
What is odd is the outline of Antarctica. Some of the big floating ice shelves (Ross, George VI, Larsen) are not included in the outline, others are (Filcher-Ronne, Amery). Theoretically, gravimetry cannot detect trends in a floating ice column, so I suggest to opt for the Ross Ice Shelf solution (include only grounded ice, i.e. the actual ice sheet).

L149: in which sense does Figure 3 serve as a reference for 1 and 2?

Figure 3: See my comment with respect to Figure 1/2.

L163-164: Please rephrase so that the “setting up” and “stacking” of equations is clear.

L164: “gravity FIELD”

L174: “In case of GRACE”; I thought this was the section about GRACE anyway.
Also, consider rephrasing “accelerations are set up”, which is unclear to me.

L188-189: Is there a reason for the poor turnout? Add a reference if possible.

L193: “data quality CORRESPONDS TO THE nominal”?

L199: “For the bad and therefore not used accelerometer observations”, consider rephrasing; “bad” is an inadequate scientific category.

L200: “stringent” --> “strict”?

Table 4: “a sigma” --> “one sigma”

L207-208: Just as in L206, give the spatial scale associated with the respective degree.

Figure 4: A colorscale is missing; the notation of “reproduced mass (degree): XY%” is not intuitive, please change. Do the “reproduced” percentages refer to the maximum mass in Greenland, or to the integrated mass? Please specify.

L222: From which value at which degree does the scaling drop to 0.01 at degree 60?

L225: What does DDK stand for, how is it set up? Please mention other non-Gaussian filters, too.

L227-237: This paragraph needs more references and possibly details. What do the authors mean by regional scaling factors? Why are classical filters not effective for low-degree solutions?

L228: “just like that”; colloquial, please rephrase.

L230: Mention that forward modelling requires additional assumptions such as mass loss being concentrated in fast flowing sections of ice streams, or at the coast, etc.

L239 onwards: It should be mentioned in this context that mass trends in Greenland are mostly related to ice loss (dynamic and melting/run-off). Also, it should be mentioned how one computes mass based on SHCs up to some cut-off degree.

L240-241: Remove this caveat? The fact that the authors’ treatment of ice mass loss is not accurate enough is implied in the following sentences.
As I understand it, the authors carry out this analysis mainly to prove that SLR and SWARM data are mostly in accordance with GRACE data. This could be stressed more clearly here instead of this caveat.

L250: What is the effect of C_20 on the mass balance of the GrIS? Possibly add a reference.

Figure 5: Consider “GRACE and SWARM gravity fields were truncated at degree 6 TO MATCH THE RESOLUTION OF SLR GRAVITY FIELDS.” or similar so that it is clear that the all fields are now consistent in spatial resolution. SLR at degree=6 is not clear just from the caption.

L255: “... to observe THE ONSET of ice LOSS”; It is not all melt, but also dynamic mass loss, with melt occurring later for icebergs drifting in the oceans, undetected by gravimetry.

L260 and L266-267: Still it might be good to at least give some possible reasons. How about orbit altitude? Also, it might be worth putting these questions in a concluding statement at the end of the paper to motivate further research.

L261-262: Why is the accuracy homogenous at low degrees?

L270: “inland ice SHEET”
Also, are accelerations, seasonal cycles, other periodicities accounted for when fitting the deterministic trends?

L281-284: I do not agree with this. It depends of course on what the authors mean by “precise”. I see that the Greenland mass loss in SLR (Figure 1, right) is located equally over South Greenland and Denmark Strait, whereas GRACE (Figure3, left) sees it along the South Greenland coast. The Antarctic mass loss is in the Amundsen Sea, i.e. completely leaked out, for SLR (Figure 2, right), and a strong gain is located over extended Enderby Land. GRACE (Figure 3, right) locates the West Antarctic loss more clearly inside the ice sheet and sees no such prominent mass gain in East Antarctica. This is of course partly due to the GRACE resolution up to degree 60. It might make sense to extend Figure 3 with GRACE at cut-off degree 6.

Figure 7: The caption is not very precise. The y-axis says “mass”. The markers are probably the mass timeseries. And the slopes of the lines are the respective trends. Please expand.

L289: “no mass trend OVER GREENLAND”?

L295: What is the order of magnitude for Greenland GIA estimates in the literature?

L300: “On the contrary” --> “In contrast to this”?

L304: straightforward

L307: Add a reference for the 1 cm and 1 mm numbers.

L309: “turns out to be”, this sounds like one would not necessarily have expected this outcome when in fact it is pretty obvious that 1/10 accuracy squared and inverted gives 100/1. Consider rephrasing.

L312-314: Please explain how decaying SWARM orbit altitude leads to higher SWARM weights, and how night-time observations in the northern hemisphere have a higher impact on seasonality than those on the southern hemisphere.

L316: What is the contribution number? Weight? Weight*Coefficient? Normalized? Please specify.

Figure 9: Same here, the weights (SWARM vs SLR) are constant in each column. Consequently, the variability per SHC stems from multiplying weight by (absolute of) SHC? Please specify.
Undefined patches (|order| > degree) should be white, or some other color off the given color scale.

L330: Mention that “sectorial” means order = degree or close.

Figure 10: It is quite hard to concentrate on the comparison of two or more graphs. Consider a different color legend to improve the visual experience.

L345: “but with higher scatter”. I cannot see this in the graph. Can you compute respective statistics of the curves and provide them here?

Figure 11: It looks like there are data points missing here.

L360: Actually, it seems that SWARM overestimates more prominently than SLR.

References

Bevis et al, 2019: Accelerating changes in ice mass within Greenland, and the ice sheet’s sensitivity to atmospheric forcing, Proc. Natl. Acad. Sci. 116(6), 1934-1939.

Peralta-Ferriz et al, 2016: Proxy representation of Arctic ocean bottom pressure variability: Bridging gaps in GRACE observations, Geophys. Res. Lett. 43, 9183-9191.

Rietbroek et al, 2014: Can GPS-Derived Surface Loading Bridge a GRACE Mission Gap?, Surv. Geophys. 35(6), 1267-1283.

Sasgen et al, 2012: Timing and origin of recent regional ice-mass loss in Greenland, Earth Planet. Sci. Lett. 333-334, 293-303.

Author Response

(The authors gave the same response as above.)

Reviewer 3 Report

The submitted study deals with the gravity field determination based on the combination of SLR data and the GRACE and SWARM missions.

Gravity field solutions for capturing the mass trends are being compared within the common period of the two satelllite missions in order to form an approach 

that could reach closer to the performance of the GRACE mission. 

In general, the manuscript is extensive and interisting.

However, the authors should respond to the following comments.

- Line 65: It is not appropriate to cite submitted manuscripts that are not published yet unless the peer-review process has been progressed successfully. Please, clarify the status.

- Line 90: "Moreover, it is important..."

Please desribe that importance rather than using this expression.

This conclusion is obviously related to the following sentence in the text. Thus, it would be good to be rephrased somehow. 

- Line 101: "..pseudo-stochastic parameterization of these orbits has to be very limited to allow for gravity field determination"

In general, estimation of pseudo-stochastic parameters may lead to overparameterisation that could potentialy absorb part of the gravity signal.

Since pseudo-stochastic parameters are applied here in the orbit modelling, could the authors give an assessment (even approximate) of these parameters' impact in the gravity field solutions and mass trends?

- Figures 1 and 2: The units sumbol 'Gt' is used widely in the amnuscript but it has not been described. Please, add the relevant descritpion i.e. Gigatons (10^12 Kgr.) 

- Lines 173-176: See the previous related commnet regarding the Line 101.

Does the constraints to zero - mentioned in line 175 - eliminate the estimation of the pseudo-stochastic parameters?

If the zero constraint is applied in order to avoid absorbing gravity signal, why the pseudo-stochastic parameters are included within the orbit determination? 

- Line 331: It would be interesting to describe more on the "weak estimation" due to the correlations.

Author Response

(The authors gave the same response as above.)

Round 2

Reviewer 2 Report

The manuscript has gone through a decisive revision, and I am happy to say that I think it is fit for being published. I’d like to thank the authors for seriously addressing the issues that I raised. The enhanced description of formalism, the presentation of not just Greenland mass trends, and the new structure will be particularly helpful and interesting for readers. I have a short list of smaller comments below, that could be addressed before publication.

Eq 1, maybe state that PHI is latitude, not co-latitude as often the case in spherical coordinates, lambda is longitude

L225: not optimal --> non-optimal

Figure 2: if true, specify that negative m implies sine-coefficients; otherwise it cannot be reconciled with Eq. 1, where m starts at 0, not at -l.

L310: Specify that ‘dimensionless SHC’ represents gravitational potential, and that the (degree-dependent) factors (Love numbers?) by Wahr translate this to surface mass density or EWH.

Antarctic coastline: I acknowledge that the procedure of integrating mass over the area shown in Figure 12 is correct, but I still advise the authors to avoid confusion by amending the coast line all the graphics to either in- or exclude all ice shelves. If they are using matlab, they might find the Antarctic Mapping Toolbox useful:

https://www.mathworks.com/matlabcentral/fileexchange/47638-antarctic-mapping-tools

or more precisely here

https://www.mathworks.com/matlabcentral/fileexchange/60246-antarctic-boundaries-grounding-line-and-masks-from-insar

Colorscales: Where the colorscale is saturated (e.g. Figure 6), it would make sense to indicate this by including respective triangles, or stating it in the caption.

L423: ‘underestimated’; in West Antarctica, ‘melt’ is an even less good description (see my comment in review round 1) because the grounded ice experiences only negligible melt (except for the Peninsula), but mostly dynamical imbalance, so I recommend to use ‘loss’ again. Also, please remind the reader that these rates are uncorrected for GIA.

Reviewer 3 Report

The authors have provided a revised manuscript addressing the points raised during the first review cycle in a satisfactory approach. 

I would therefore recommend the paper for publication to Remote Sensing MDPI in its current version.

However, prior the final version to be published please check the status of the cited manuscript and edit accordingly as per the first comment of my first review.